# Effects of Amendments on Physicochemical Properties and Respiration Rate of Soil from the Arid Region of Northwest China

**Dianpeng Li** [1,2] , **Jianqin Zhou** [2] , **Yuxin Zhang** [2] , **Tao Sun** [2] , **Shuqing An** [1,3],* and **Hongtao Jia** [2],*

1 Institute of Wetland Ecology, School of Life Science, Nanjing University, Nanjing 210093, China; lldp0628@gmail.com

2 College of Grassland and Environment Sciences, Xinjiang Agricultural University, Urumqi 830052, China; machine1229@163.com (J.Z.); zyx220171071@163.com (Y.Z.); stao1211@163.com (T.S.)

3 Nanjing University Ecological Research Institute of Changshu, Nanjing University Research Institute (Changshu) Co., Ltd., Suzhou 215501, China

\* Correspondence: anshq@nju.edu.cn (S.A.); jht@xjau.edu.cn (H.J.)

**Abstract:** In arid regions, decreased soil fertility has adversely affected agricultural sustainability. The effects of different amendments in alleviating these issues and increasing soil fertility remain unclear. Herein, a two-year field experiment was conducted to evaluate the properties of grey desert soil and soil respiration (SR) dynamics under six different treatment groups: biochar (BC), leonardite (LD), anionic polyacrylamide ($PAM^-$), cationic polyacrylamide ($PAM^+$) powder, anionic polyacrylamide solution in water ($PAM^-W$), and control (CK). We observed that the BC and LD amendments significantly altered soil pH, organic matter, available nitrogen, available phosphorus, cation exchange capacity, and SR. PAM amendment increased the SR as compared to the control, except in autumn, but PAM did not affect the soil properties. SR under different amendments showed strong seasonal patterns, the highest and lowest SR rates were observed in June and January, respectively. Amendments and seasonal dynamics significantly affected SR, but no interaction was observed between these factors. Temporal variation of SR was substantially influenced by soil temperature at 15 cm of soil depth. Temperature sensitivity of SR ($Q_{10}$) increased with soil depth and decreased with amendment addition. SR was significantly affected by soil temperature, moisture, air temperature, and their interactions. The outcomes of this study suggested that the BC and LD amendments improved soil fertility and negated the net carbon accumulation by increasing the SR and $Q_{10}$ in arid agriculture soil.

**Keywords:** soil conditioner; grey desert soil; $CO_2$ emission; soil fertility; $Q_{10}$

## 1. Introduction

Grey desert soil represents an important soil type in the typical continental arid climate regions of northwestern China, out of the total 52,528 km² of cultivated grey desert soil across China, one quarter is cultivated in Xinjiang [1]. Due to a high evaporation rate and agriculture practice of using high-intensity mechanical operations in the arid climate, soil in the arid region encounters secondary salinization and the tight barrier layer. This hampers the utilization rate of water and fertilizer as well as the productivity of the soil [2,3]. In addition, in fields where continuous farming was carried out for a duration of 5–10 years, activities of catalase, invertase, and protease in grey desert soil decreased notably [4]. These obstacles have severely restricted crop growth and adversely impacted crop yield [5,6].

In recent years, soil amendments have been reported to be a highly effective approach for increasing the fertility and microbial activity of the soil, particularly in low-fertile soil of arid region [7–9]. Biochar (BC), a carbon-rich amendment, has received wide attention in agriculture research areas, it can sequester carbon and obtund climate change, impeding soil nutrients' loss caused by leaching and improving soil quality [10,11]. Biochar

amendment can substantially increase organic matter, nutrients, microbial activity and decrease bulk density and soil aggregation [12,13]. Leonardite (LD), a waste product of coal production that is commonly present on the surface of coal, is widely distributed across Xinjiang at Hutubi County, Shanxi (Luliang City), and Inner Mongolia (Ordos) provinces in China. Humic acid is the main component of LD, and it contains a high level of aromatic and acidic functional groups [14]. As per the previous study, humic acid can serve as an effective amendment for soil [14]. LD amendment increases available soil nutrients, alters the soil's ionic composition, and reduces $Na^+$ toxicity in soil [15,16]. Polyacrylamide (PAM), a synthetic soil conditioner, has been used since the 1990s to reduce soil erosion and enhance infiltration. Commonly used PAM ion types are anionic ($PAM^-$) and cationic ($PAM^+$), and the adverse effects of $PAM^-$ on aquatic macrofauna, edaphic micro-organisms, or crop species have not been documented [17]. Natural Resources Conservation Service (NRCS) has specified $PAM^-$ application for controlling irrigation-induced erosion [17]. $PAM^-$ amendment to the soil restricts nitrogen accumulation in soil and surface runoff of nutrients [18]. Previous studies have shown that $PAM^-$ treatment could improve the stability of soil aggregates [19]. As observed in multiple studies, $PAM^-$ application affects surface runoff of nutrients and reduces soil erosion; besides, it affects nutrient cycles and microbial activities in the soil [20]. Previous studies have rarely explored the application methods for $PAM^+$ [19,21] and how these amendments' addition alter the grey desert soil's properties. In addition, these amendments' effect on SR in the arid region remains elusive.

Due to the drop irrigation and fertilization practices, the SR rate of the arid region cropland was 2–5 times higher than the natural ecosystem [22]. The annual SR of farmland soil accounts for 10% of the soil carbon emissions (66.62–100.72 Pg C) [23,24]. BC, when applied as an amendment, can stimulate [25], inhibit [26], or has no effects on SR [13,27]. For instance, the BC application did not increase SR in the semi-arid soil [28]. Moreover, BC rapidly increased sandy clay loam soil SR in the first few hours or days of application [29,30], but it did not affect soil respiration in agricultural soil types across China [27]. LD application to soil altered the diurnal variation pattern of SR and significantly increased SR rate by 17.1% to 35.2% [16]. Matsuoka et al. [31] and Wen et al. [32] reported that soil microbial culture could utilize PAM as a nitrogen and carbon source, and $PAM^-$ can stimulate nitrification and carbon mineralization [21]. In addition, as per other studies, $PAM^-$ did not increase the decomposition of native or added carbon in soils [25]. $Q_{10}$ is an important parameter to evaluate the feedback intensity between $CO_2$ emission and global warming [33]. As amendments remarkably affect the soil carbon content, enzymatic activities, and microbial communities, biochar could also affect the $Q_{10}$. SR in biochar-amended soils had a higher $Q_{10}$ than soil without biochar amendment [34,35]. LD application to soil did not affect $Q_{10}$ [16], but only a few studies have explored the effect of PAM on $Q_{10}$. In summary, multiple studies have explored the effects of soil amendment application, specifically biochar amendment application, on basic soil properties, SR and $Q_{10}$. However, the information about how amendments alter the grey desert SR and $Q_{10}$ remains unexplored.

Herein, we examined the effects of BC, LD, and PAMs amendments on basic properties of soil and SR rate in cropland grey desert soil. In this study, a two-year in situ experiment was designed to answer the following question: (a) how do amendments addition affect soil properties? (b) how do SR and its temperature sensitivity change with different amendments?

## 2. Materials and Methods

### 2.1. Study Site

In situ experiments were performed in Urumqi (87°33′47.44″ E, 43°48′42.63″ N) with the mean annual precipitation and temperature of 260 mm and 8.6 °C, respectively. This area has a temperate continental arid climate. The study site contained grey desert soil as per the Chinese soil classification system and calcareous desert soil as per the FAO soil classification [36]. In the upper 20 cm soil layer, the percentage of sand (0.02–2 mm), silt (0.002–0.02 mm), and clay (<0.002 mm) were found to be 33.1%, 53.7%, and 13.2%, respectively.



The soil bulk density was 1.41 g/cm$^3$, and other properties of the soil are shown in Table 1. Primary vegetation in this soil was comprised of Chenopodiaceae (*Anabasis*), Gramineae (*Stipa sareptana Becher*), and Compositae (*Artemisia*) before application of soil amendment.

**Table 1.** Basic physicochemical properties of soil and amendments.

| Soil & Amendments | pH | EC (dS/cm) | OC | TN | TP | CEC (cmol+/kg) |
|---|---|---|---|---|---|---|
| | | | | (g/kg) | | |
| Soil | 8.62 ± 0.22 [b] | 0.22 ± 0.02 [c] | 7.41 ± 0.09 [b] | 0.78 ± 0.07 [d] | 0.74 ± 0.04 [b] | 3.84 ± 0.26 [b] |
| Biochar | 9.37 ± 0.11 [a] | 3.70 ± 0.19 [a] | 417 ± 9 [a] | 21.8 ± 0.9 [a] | 10.6 ± 0.3 [a] | 12.0 ± 0.32 [a] |
| Leonardite | 4.87 ± 0.26 [d] | 1.33 ± 0.21 [b] | 431 ± 11 [a] | 8.70 ± 0.1 [c] | 0.24 ± 0.05 [c] | |
| Anionic PAM | 7.36 ± 0.09 [c] | | 434 ± 15 [a] | 16.2 ± 0.4 [b] | | |
| Cationic PAM | 7.06 ± 0.12 [c] | | 421 ± 8 [a] | 16.2 ± 0.3 [b] | | |

Note: EC, electric conductivity; OC, organic carbon; TN, total nitrogen; TP, total phosphorous; CEC, cation exchange capacity, different letters in columns indicate significant differences ($p < 0.05$).

### 2.2. Soil Amendments

The BC used in this study was produced using a cotton stalk, where cotton was pyrolyzed with a residence time of 2 h at 400 °C in a muffle furnace and later grounded to powder. The powdered biochar was passed through a 2 mm sieve to obtain the final product. LD was collected from Hutubi county, Xinjiang Uygur Autonomous Region (43°47′ N, 86°31′ E). PAM was purchased from Henan Jiechuang Water Treatment Material Co. Ltd. (Zhengzhou, Henan, China). PAMs was white powder with [C$_3$H$_5$ON]$_n$ molecular formula, 12 million Da molecular weight, and 20% hydrolysis. The basic properties of BC, LD, PAM$^-$, and PAM$^+$ are shown in Table 1.

### 2.3. Experiment Regimes

To explore the effects of soil amendments on soil properties and respiration in grey desert soil, we conducted a soil amendment experiment in April 2015. Before initiating the soil spatial heterogeneity reduction experiment, the 0–20 cm soil layer was thoroughly mixed. The application rates of BC, LD, and PAMs were based on a previous study by Streubel et al. [37], Li et al. [38], and Wang et al. [18]. These experiments included six different treatments: 2.0 kg biochar/m$^2$ (BC), 2.7 kg leonardite/m$^2$ (LD), 1.0 g anionic polyacrylamide/m$^2$ (PAM$^-$), 1.0 g cationic polyacrylamide/m$^2$ (PAM$^+$), and 1.0 g anionic polyacrylamide/m$^2$ in water or (PAM$^-$W), and without any amendment application (CK). These amendments were applied in the first year of the experiment. To minimize the impact of plant–soil microbe interaction and respiration, crops and plants were excluded from all six treatments. To remove any plantation from the study site, all the seedlings were pulled. In March 2015, eighteen 2 m × 2 m plots were separated by a 0.5 m buffer and arranged following a randomized block design at the experimental site. Soil amendments were spread on the soil surface of the BC, LD, PAM$^-$, and PAM$^+$ treatment plots. BC, LD, and PAM were incorporated into the topsoil (20 cm depth) by plowing and leveling with a rake, and later soil was watered using 2 L pure water. For PAM$^-$W treatment, 2.0 g cationic PAM was mixed with 2 L water and then applied to 2 m × 2 m plot. Two PVC collars with 20 cm diameter and 10 cm height were placed 5 cm above the ground level in each plot, and with a 0.5 cm distance, collars were placed at a 0.5 m distance from the plot edge.

### 2.4. Soil Temperature, Moisture, and SR Measurements

Soil temperature at 5 (ST5), 10 (ST10), and 15 (ST15) cm depth were measured using 6 groups of geothermometers (Wuqiang Inc., Hengshui, Hebei, China). The air temperature (AT) of the study site was measured using an air thermometer. To determine the soil moisture (SM), soil samples were collected using stainless steel corer, and soil from 0 to 20 cm depth was collected monthly. Soil from each treatment was sampled at a 20 cm distance from PVC collars. Fresh soil samples were oven-dried for 24 h at 105 °C. We were not able to collect soil moisture data from November to February 2015–2016 and 2016–2017, as the

soil froze, and it was difficult to sample frozen soil. The experimental site was covered with snow from 21 November 2015 to 16 March 2016 and 3 December 2016 to 19 March 2017. The snow thickness was 0.2–6.7 cm. In addition, the snow thickness exceeded the PVC collar's height during December 2015 and December 2016. SR was measured from April 2015 to May 2017, after watering the plots for 3 days and using LI-8100A Automated Soil $CO_2$ Flux System (LI-COR Inc., Lincoln, NE, USA). SR was determined daily at 09:00 and 12:00 [39]. A 100 s period of $CO_2$ concentration in the chamber was measured and recorded every second by the system in a closed chamber. The litter and new plants were removed from each PVC collars and plot before measuring SR.

### 2.5. Measurement of Soil and Amendment Properties

Before initiating the experiment, the soil samples were collected from the experimental area from the diagonal direction to measure the basic soil properties as shown in Section 2.1 and Table 1. After recording the last SR measurement (May 2017), PVC collars were removed, and soil from 0 to 20 cm depth was sampled from each plot and then transported to the laboratory to analyze the basic properties of the soil. The methods for determining the basic properties of soil and amendments are as follows: soil bulk density (BD) at 20 cm depth was measured gravimetrically at 105 °C for 48 h; pH and electric conductivity (EC) of soil, biochar, and leonardite were measured in a 1:2.5 solution of soil, biochar, leonardite in water, respectively, using a Metter-Toledo 320 pH meter and electric conductivity meter, respectively (FE-30, Shanghai, China); soil organic carbon/matter (OC/SOM) was measured using wet oxidation with $K_2CrO_7$ in sulfuric acid at 170–180 °C; soil total carbon (TC) and total nitrogen (TN) were measured with the elemental analyzer (Euro EA3000-Single, EuroVector, Milan, Italy); soil available nitrogen (AN) was detected using the alkaline hydrolysis diffusion method; soil total phosphorus (TP) concentration was digest by $H_2SO_4$–$HClO_4$ and then measured using Mo–Sb colorimetry method; the available phosphorus (AP) was extracted with 0.5 mol/L. $NaHCO_3$ and measured using a UV-1800 spectrophotometer (Shimadzu, Kyoto, Japan); soil cation exchange capacity (CEC) was determined by using sodium acetate (1 mol/L, 33 mL for 4 g soil) and flame photometer method [40], as shown in Table 1. The pH, EC, TN, TP, and CEC of biochar were significantly higher than the leonardite, $PAM^-$, and $PAM^+$.

### 2.6. Statistical Analysis

Analysis of variance (ANOVA) was used to determine the differences between mean SR among seasons and different treatments. Responses of SR to amendments, sample time (month), and their interaction was analyzed through two-way ANOVA. We used linear and non-linear models to quantify the correlation between environmental factors and SR ($\mu mol/m^2/s$). Non-linear regression analysis of SR against ST [41], and linear regression analysis of SR against AT (°C), ST (°C), and SM (%) was performed using the following equations:

$$SR = ae^{bST} \tag{1}$$

$$Q_{10} = e^{10b} \tag{2}$$

$$SR = X \times c + Y \times d + Z \times e + f \tag{3}$$

where a and f are fitted parameters, $Q_{10}$ is the temperature sensitivity of SR, X, Y, and Z were environmental factors, such as ST, AT, and SM.

We used quadratic function model to determine the relationship between SR ($\mu mol/m^2/s$) and SM (%) [41]:

$$SR = g + h \times SM + m \times SM^2 \tag{4}$$

where g, h, and m are fitted parameters.

## 3. Results

### 3.1. Effect of Soil Amendments on Soil Properties

BC and LD amendments significantly increased EC, SOM, AN, AP, and CEC as compared to CK but did not affect TN, TP, and BD (Table 2). The pH, AN, AP, and CEC in BC treatment were significantly higher than LD treatment ($p < 0.05$), while LD had the highest EC. However, there was no significant difference between SOM, TN, TP, and BD in BC and LD treatments ($p > 0.05$). The PAM$^-$, PAM$^+$, and PAM$^-$W amendments did not affect soil properties, except EC (Table 2).

**Table 2.** Effects of amendments on soil properties.

| Treatment | pH | EC (μS/cm) | SOM (g/kg) | TN (g/kg) | TP (g/kg) | AN (mg/k g) | AP (mg/k) | CEC (cmol₊/kg) | BD (g/cm³) |
|---|---|---|---|---|---|---|---|---|---|
| CK | 8.63 ± 0.04 [b] | 242 ± 8 [c] | 12.53 ± 0.03 [b] | 0.727 ± 0.020 [a] | 0.695 ± 0.009 [a] | 29.97 ± 1.86 [c] | 6.93 ± 0.12 [c] | 3.88 ± 0.05 [c] | 1.41 ± 0.01 [a] |
| BC | 8.89 ± 0.02 [a] | 253 ± 4 [b] | 12.66 ± 0.05 [a] | 0.733 ± 0.005 [a] | 0.705 ± 0.008 [a] | 41.93 ± 1.68 [a] | 9.40 ± 0.16 [a] | 4.26 ± 0.08 [a] | 1.42 ± 0.02 [a] |
| LD | 8.08 ± 0.04 [c] | 267 ± 3 [a] | 12.64 ± 0.07 [a] | 0.735 ± 0.009 [a] | 0.698 ± 0.017 [a] | 37.03 ± 0.17 [b] | 8.41 ± 0.08 [b] | 4.07 ± 0.07 [b] | 1.42 ± 0.00 [a] |
| PAM$^-$ | 8.60 ± 0.02 [b] | 250 ± 3 [b] | 12.57 ± 0.04 [ab] | 0.727 ± 0.012 [a] | 0.720 ± 0.020 [a] | 32.57 ± 0.53 [c] | 7.04 ± 0.07 [c] | 3.85 ± 0.03 [c] | 1.42 ± 0.02 [a] |
| PAM$^+$ | 8.58 ± 0.02 [b] | 248 ± 2 [b] | 12.61 ± 0.03 [ab] | 0.725 ± 0.011 [a] | 0.709 ± 0.011 [a] | 31.73 ± 1.41 [c] | 6.91 ± 0.15 [c] | 3.86 ± 0.03 [c] | 1.41 ± 0.00 [a] |
| PAM$^-$W | 8.59 ± 0.03 [b] | 247 ± 3 [b] | 12.59 ± 0.06 [ab] | 0.720 ± 0.008 [a] | 0.704 ± 0.008 [a] | 32.43 ± 0.50 [c] | 7.04 ± 0.08 [c] | 3.88 ± 0.04 [c] | 1.41 ± 0.02 [a] |

Note: CK, control; BC, biochar; LD, leonardite; PAM$^-$, anionic polyacrylamide; PAM$^+$, cationic polyacrylamide; PAM$^-$W, anionic polyacrylamide application with water; data (mean, n = 3) within the same properties followed by different letters in columns indicate significant differences ($p < 0.05$). EC, electric conductivity; SOM, soil organic matter; TN, total nitrogen; AN, available nitrogen; TP, total phosphorous; AP, available phosphorus; CEC, cation exchange capacity; BD, bulk density.

Soil temperature at different depths exhibited clear seasonal patterns. The mean soil temperature at 5, 10, and 15 cm soil depth were 17.07 °C, 13.26 °C, and 10.09 °C, respectively, and the mean air temperature was 14.50 °C (Figure 1a,b). Mean soil moisture content under CK, BC, LD, PAM$^-$, PAM$^+$, and PAM$^-$W amendments were 12.64%, 12.86%, 13.67%, 14.09%, 14.06%, and 14.56%, respectively (Figure 1c). However, no significant differences were observed between different amendments.

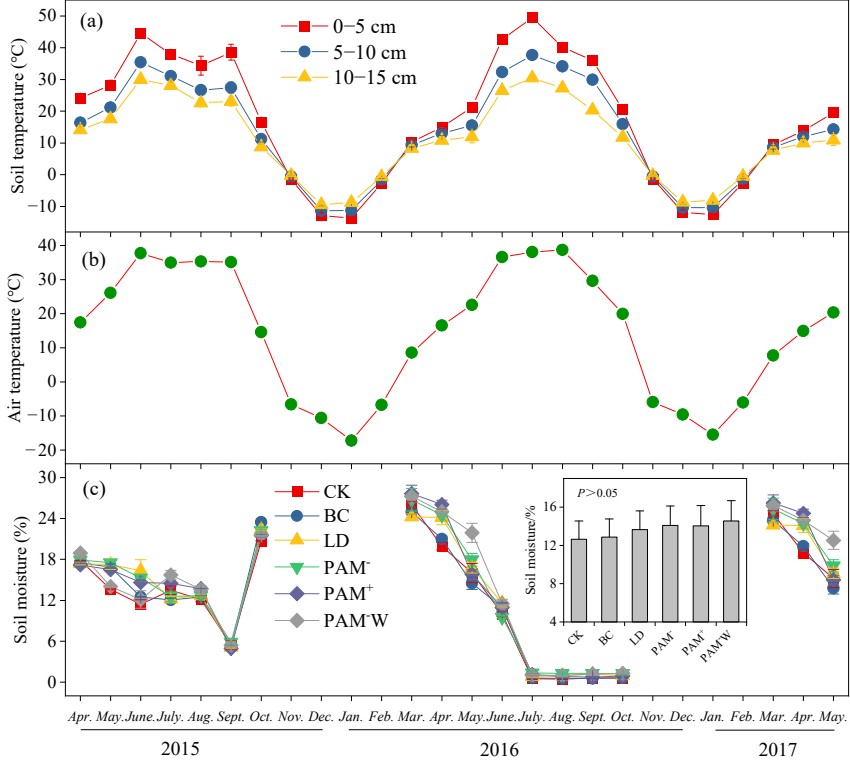

**Figure 1.** Dynamics of (**a**) soil temperature, (**b**) air temperature, and (**c**) soil moisture during 2015 to 2017. CK, control; BC, biochar; LD, leonardite; PAM$^-$, anionic polyacrylamide; PAM$^+$, cationic polyacrylamide; PAM$^-$W, anionic polyacrylamide application with water.

### 3.2. Dynamics of SR

SR exhibited clear seasonal patterns in different amendments. The highest and lowest respiration rates of all treatments were observed in June and January, respectively; however, no significant differences were observed between the first year and second year of the study (Figure 2).

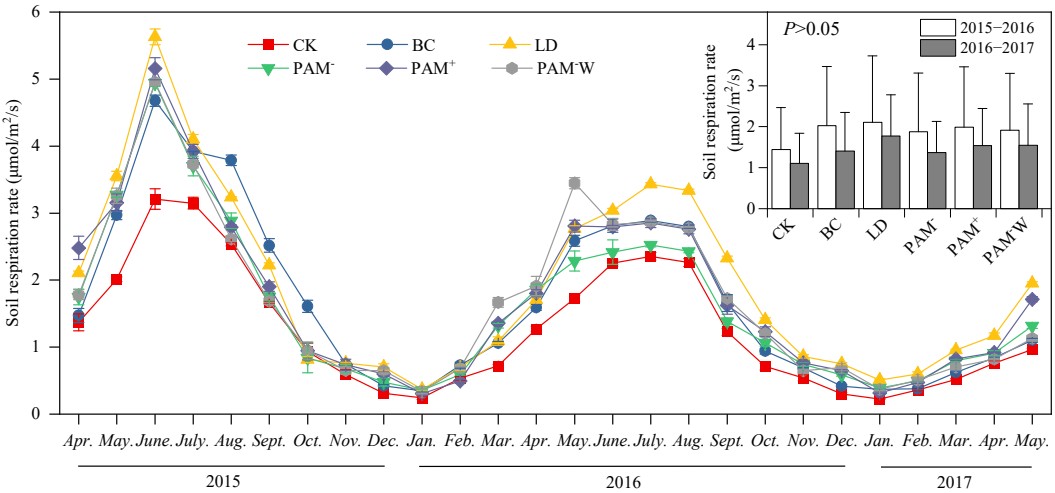

**Figure 2.** Dynamics of SR from 2015 to 2017. The codes of CK, BC, LD, PAM$^-$, PAM$^+$, and PAM$^-$W are same as those in Figure 1.

SR rate under BC and LD amendment was significantly higher than CK. Irrespective of soil amendments, a total soil respiration of 44.3% was observed during summer. The seasonal order of SR flux magnitudes was as follows: summer > spring > autumn > winter (Figure 3a). The summer SR rate was significantly higher than spring, autumn, and winter. In different seasons, SR did not vary significantly in different amendments. BC treatment increased SR in summer and autumn compared to CK, while LD treatment notably increased SR in all seasons. PAM$^-$ amendment significantly increased SR in spring and winter compared to CK; PAM$^+$ and PAM$^-$W significantly increased SR, except in autumn; but PAM type and application method did not affect SR (Figure 3b). During the experimental period, BC and LD amendments significantly increased SR rate compared to CK, but no significant differences were observed between different amendments (Figure 3b). Moreover, SR was significantly affected by soil amendments each month ($p < 0.05$), but no interactions were observed between these two factors (Table 3).

**Table 3.** Responses of SR to amendment, sample time (month), and their interaction in two-way ANOVA.

| Parameters | Soil Respiration | | |
|---|---|---|---|
| | DF | F | p |
| Amendment | 5 | 3.017 * | 0.015 |
| Month | 11 | 43.005 ** | <0.001 |
| Amendment × Month | 55 | 0.211 | >0.05 |

Note: * $p < 0.05$, ** $p < 0.01$.

### 3.3. Driving Factors for SR

SR was exponentially correlated to ST at different soil depths (Figure 4). Adjusted $R^2$ values indicated that the ST5, ST10, and ST15 accounted for 79.3%, 80.3%, and 87.0% of the SR variations, respectively. The $Q_{10}$ value for each treatment increased with the soil depth. $Q_{10}$ at 5, 10, and 15 cm were 1.36, 1.50, and 1.67, respectively. In addition, the $Q_{10}$ at 15 cm was significantly higher than $Q_{10}$ at 5 and 10 cm. Different soil amendments decreased the $Q_{10}$ value (Figure 5) as compared to CK.

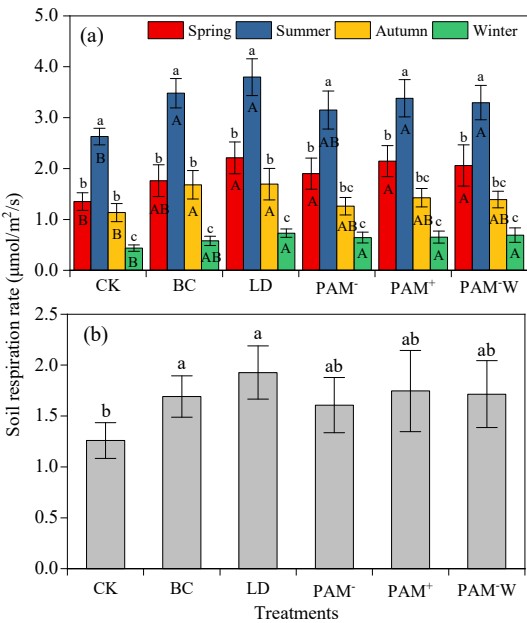

**Figure 3.** (**a**) Average SR rate in different seasons and (**b**) mean value for the different amendment treatments during the experimental period (April 2015 to May 2017). In Figure 3a, the same treatment in the different seasons followed with different lowercase letters represent a significant difference (*p* < 0.01) based on the LSD test. Different treatments in the same season followed with different uppercase letters represent a significant difference (*p* < 0.01) based on the LSD test. The codes of CK, BC, LD, PAM⁻, PAM⁺, and PAM⁻W are same as those in Figure 1.

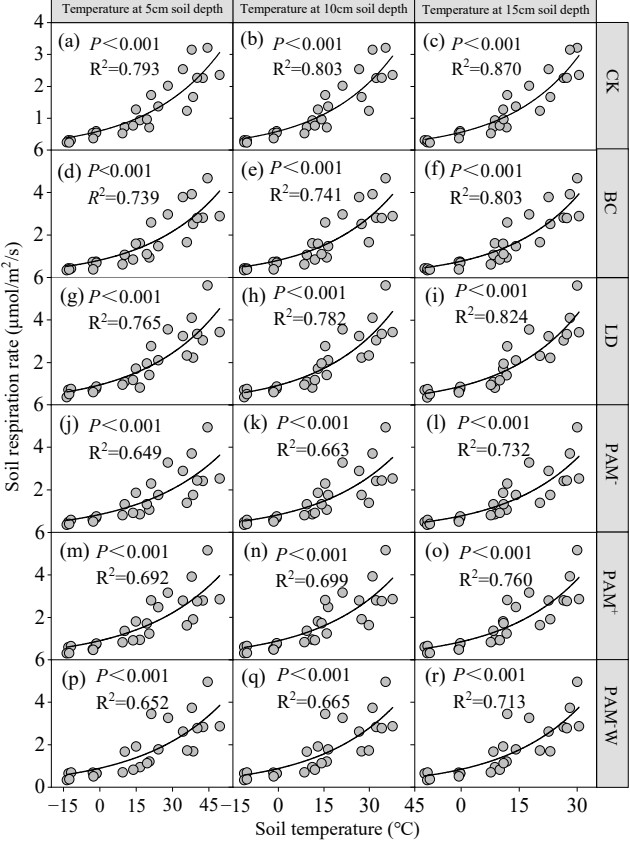

**Figure 4.** The correlation between SR and soil temperature at different soil depths. The codes of CK, BC, LD, PAM⁻, PAM⁺, and PAM⁻W are same as those in Figure 1.

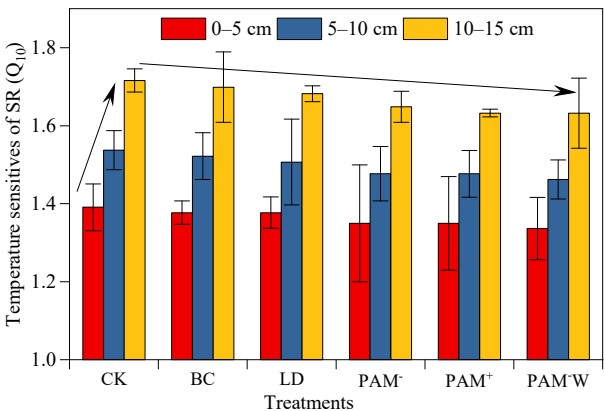

**Figure 5.** Temperature sensitivities of SR ($Q_{10}$) under different treatments. The codes of CK, BC, LD, PAM$^-$, PAM$^+$, and PAM$^-$W are same as those in Figure 1.

SR was positively and non-linearly correlated to SM (Figure 6). Adjusted $R^2$ values indicated that the SM level accounted for up to 45% of SR variations. ST, AT, SM, and their interactions significantly influenced SR. For CK and BC treatment, the two-variable model could be used to explain the correlation of SR to AT and SM with adjusted $R^2$ ranging from 0.873 to 0.896 and significant regressions ($p < 0.0001$). The SR of LD, the one-variable linear model, could be used to describe the correlation between SR and ST with 0.722 $R^2$. SR of PAMs could be predicted by ST and SM. The equations explained 64.4–74.6% of the variation in SR in our data (Table 4).

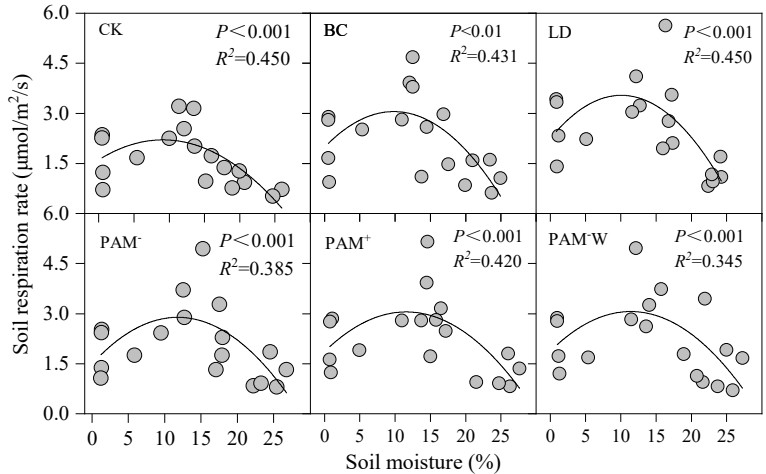

**Figure 6.** Correlation between SR and soil moisture (SM) under different soil amendments. The codes of CK, BC, LD, PAM$^-$, PAM$^+$, and PAM$^-$W are same as those in Figure 1.

**Table 4.** Linear mixed-effect model of SR and soil temperature (ST), air temperature (AT), and soil moisture (SM) for different treatments.

| Treatment | Regression Equation | $R^2$ | F | *p*-Value |
|---|---|---|---|---|
| CK | $SR = SM \times 0.060 + AT \times 0.103 - 1.715$ | 0.896 | 64.406 | <0.0001 |
| BC | $SR = SM \times 0.088 + AT \times 0.142 - 2.509$ | 0.873 | 51.429 | <0.0001 |
| LD | $SR = ST15 \times 0.129 + 0.212$ | 0.722 | 44.204 | <0.0001 |
| PAM$^-$ | $SR = ST15 \times 0.153 + SM \times 0.071 - 1.641$ | 0.746 | 22.058 | <0.0001 |
| PAM$^+$ | $SR = ST15 \times 0.142 + SM \times 0.050 - 0.956$ | 0.694 | 16.976 | <0.0001 |
| PAM$^-$W | $SR = ST15 \times 0.147 + SM \times 0.057 - 1.229$ | 0.644 | 13.557 | <0.0001 |

Note: The codes of CK, BC, LD, PAM$^-$, PAM$^+$, and PAM$^-$W are same as those in Table 2.

## 4. Discussion

### 4.1. Effects of Soil Amendments on SR

In the present study, we observed that BC and LD amendment to arid agriculture soil affected its physicochemical properties and SR as compared to control. BC exhibited its effects on SR through three different processes. Firstly, BC amendment increased the labile exogenous organic carbon in the soil. Labile fractions of BC are an important source of carbon that is used as a selective substrate in microbial activity [42]. Smith et al. [30] suggested that soil $CO_2$ is primarily contributed by the decomposition of labile carbon fractions of the BC. Secondly, the BC amendment provides better environmental habitat for soil microbes, increasing the soil microbial activity and organic carbon decomposition [43]. Thirdly, the BC amendment improves the soil's physical structure and enhances soil's air permeability, increasing the SR [44,45]. In this study, BC amendment increased SOM and SR. It indicated that BC negated the net carbon accumulation by increasing SR in grey desert cropland soil.

From previous studies, LD amendment exhibited its effect on SR from two aspects. One side, the high humic acid content and porous structure of LD improved the soil's environmental habitat, altering the water–heat exchange status of soil and affecting soil enzyme activities [38]. On the other side, phenolic hydroxyls, hydroxyl groups, ion exchange, and complexation in LD increased the organic carbon mineralization rate [46]. The slight stimulatory effects of PAM on SR can be discussed in two ways: (a) PAM amendment significantly decreases the soil pH value and alters the soil chemical properties, stimulating microbial biomass growth and respiration [31] and (b) PAM acts as the substrate for microbes [20] in culture medium or soil. *Bacillus sphaericus*, *Acinetobacter*, *Bacillus cereus*, and *Bacillus flexus* strains can break down the PAM and utilize it as the single source of carbon and nitrogen, which results in increased microbial growth [32]. Thus, the PAM improves soil condition and acts as a substrate, affecting SR. PAMs application stimulated the SR in spring, summer (except PAM$^-$), and winter. Similar findings were reported by Watson et al. [20]. This study showed that PAM significantly increased SR as compared to control. Matsuoka et al. [31] demonstrated that *Bacillus sphaericus* and *Acinetobacter* strains could utilize PAM as the exclusive nitrogen and carbon source in the culture medium. A previous study demonstrated that the addition of wheat stubble to PAM enhanced PAM breakdown by *Basidiomycetes*. This study showed that basidiomycetes could enzymatically degrade a myriad of nonphenolic complex substrates [47]. It indicated that although the PAM was highly resistant to microbial degradation, its breakdown could increase the $CO_2$ evolved from the microbial biomass. This study also demonstrated that the application methods and ion type of PAM have no effect on SR.

SR followed similar seasonal patterns for all other amendments (Figure 2). Similar results have been observed in a typical karst calcareous cropland soil, hot–wet environmental conditions increased the microbial decomposition of organic matter, resulting in an increased SR rate [48]. In the present study, amendments addition and month had a significant effect on SR, and no interactions between the two factors were observed (Table 2). Soil temperature is the key factor that regulates the seasonal variation in soil microbial biomass, and thus, it controls SR [49]. In an environment with appropriate soil moisture, at 15 cm of soil depth, the temperature was one of the major factors that regulated the temporal variation in SR by influencing soil enzymes and microbial activities [50]. It indicated that high SR variability at high temperature is probably the result of variable soil moisture, and lower SR variability at low temperature is probably the result of lower water content.

### 4.2. Amendments Decrease Temperature Sensitivity of SR ($Q_{10}$)

$Q_{10}$ is a crucial parameter for predicting the fate of soil carbon under global warming [51]. In the present study, $Q_{10}$ was strongly correlated to the ST depth under different amendments; besides, it increased from 1.36 to 1.67. Xu et al. [52] showed similar results in the Qinghai–Tibetan plateau in China. As soil stable carbon fraction increases with increasing soil depth, it is difficult for soil microbes to utilize deep stable organic car-

bon. This is one of the reasons that explain a higher $Q_{10}$ value at a higher soil depth [53]. Khomik et al. [54] showed that $Q_{10}$ ranged from 3.6 to 12.7 at temperature in a 2~50 cm soil depth. The highest $Q_{10}$ was observed at a 10 cm soil depth. Peng et al. [55] reported that $Q_{10}$ at a 0 cm soil depth was half to that of $Q_{10}$ at a 20 cm soil depth. According to a study by Pavelka et al. [56], grassland ecosystems have the most appropriate surface temperature. Thus, we concluded that the best temperature to predict $Q_{10}$ in the different ecosystems might vary. Raich and Schlesinger [57] reported a $Q_{10}$ of 2.4 on the global scale. The average $Q_{10}$ values of the different ecosystems in China ranged from 1.81 to 3.05 and 2.25 in the cropland [53]. However, these values were much lower in the current study. It indicated that the temperature sensitivity of SR in the arid region cropland was lower than in another ecosystem. Different amendments decreased the $Q_{10}$ at different soil depths as compared to CK. It might be due to the high content of exogenous organic carbon in BC and LD amendments. It contributed to a high level of aromatic substances and thus increased the stability of the organic carbon pool and decreased the $Q_{10}$. $Q_{10}$ in different amendments varied due to diminutive changes in SOM. It significantly impacted substrate availability for microbes as compared to CK, thereby affecting the $Q_{10}$ value [52,58].

*4.3. Effect of Amendments Addition on Soil Properties*

In this study, BC amendment significantly increased soil pH value. Previous studies have reported similar findings with the application of different BC amendments to different soil types. BC amendment increased soil pH from 0.19 to 3.47 units [59]. It might be due to the alkaline pH of BC (pH = 9.37), negatively charged phenolic, carboxyl, and hydroxyl groups on BC surfaces [60], and the $H^+$ in soil solution [61]. In the current study, LD amendment significantly decreased soil pH as compared to control (Table 1). Previous studies have shown that soil pH decreased under the LD amendment [62]. In this study, LD amendment decreased soil pH, which might be related to its relatively higher acidity (pH = 4.87). Additionally, as the exogenous carbon source, LD stimulated the microbial decomposition of organic carbon and exudate excretion (i.e., formate, amino, and enzymes) with decreasing soil pH [63]. BC and LD application increased soil CEC value as compared to CK. Liang et al. [64] demonstrated that CEC increase could be attributed to the high surface area and charge density of the BC. In addition, oxidation of aromatic carbon on the BC surface to carboxylic groups increased the CEC [65,66]. Moreover, distinct PAMs and amendment application methods did not affect the soil's physicochemical properties since the decomposition rate of PAM was about 10% per year [67,68].

BC and LD application to soil significantly increased SOM, AN, and AP. It might be due to a higher level of organic carbon in BC and LD and a higher level of humic acid-type substances in LD [38]. In addition, BC addition was favorable for the humus formation through adsorption and aggregation [69]. We did not observe any significant differences between the three PAM treatments, which might be due to the stable physicochemical properties of PAM [67]. Previous studies have reported that BC amendment decreased soil BD and improved soil quality [70]. However, in the current study, amendments did not show any effect on soil bulk density. Thus, the long-term effects of BC on soil physicochemical properties demand further investigation.

## 5. Conclusions

This study demonstrated that the BC and LD amendments significantly altered the soil nutrients and SR in grey desert soil, and the highest SR was recorded in summer. SR was significantly and non-linearly correlated to ST and SM. $Q_{10}$ at 15 cm was higher than $Q_{10}$ at 5 cm and 10 cm. Soil temperature, air temperature, and soil moisture and their interaction significantly influenced SR of amended soil. PAMs treatment did not affect soil properties significantly but promoted SR in spring, summer (except $PAM^-$), and winter. BC and LD amendment increased SR and SOM but decreased the $Q_{10}$. Therefore, amendment (BC and LD) application will negate the net carbon accumulation by increasing SR and decreasing $Q_{10}$ in grey desert soil.

**Author Contributions:** Conceptualization, D.L. and H.J.; methodology, D.L. and Y.Z.; formal analysis, J.Z.; writing—original draft preparation, D.L.; Investigation, Y.Z. and T.S.; writing—review and editing, S.A., T.S., and H.J.; and visualization, S.A. and J.Z. All authors have read and agreed to the published version of the manuscript.

**Funding:** This work was funded by the National Natural Science Foundation of China (31560171), Water Resources Science and Technology Project of Jiangsu Province, China (No. 2018064), and the Major Science and Technology Program for Water Pollution Control and Treatment, China (No. 2017ZX07602-002).

**Institutional Review Board Statement:** Not applicable.

**Informed Consent Statement:** Not applicable.

**Data Availability Statement:** The data presented in this study are available on request from the corresponding author.

**Acknowledgments:** The authors would like to thank all the reviewers who participated in the review and MJEditor (www.mjeditor.com, accessed on 11 March 2021) for its linguistic assistance during the preparation of this manuscript.

**Conflicts of Interest:** The authors declare no conflict of interest.

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
