# Peer review of "Effects of Amendments on Physicochemical Properties and Respiration Rate of Soil from the Arid Region of Northwest China"

_sustainability, doi:10.3390/su13105332_

Round 1
Reviewer 1 Report
General remarks: This is a well designed and organised work with detailed evaluation of results of a soil amendments test. I have only some remarks and suggestions to this work.
Abstract
Line 21: “PAM amendment did not affect soil properties and SR.” Based on Fig. 3 a), different PAM treatments increased the SR comparing to the control, except in autumn.
Introduction
I suggest to add some sentences about soil respiration, as the main studied parameter of the work.
Materials and methods
Line 129: Please, change “electronic” to electric.
Line 163: “Basic physicochemical properties of soil and amendments”
Results
Lines 170-173: Comparing the effects of BC and LD is a little bit confuse. First you mentioned no differences between them, after that difference in AN level was written, while in Table 2 AP and AN results are different between these two treatments.
Lines 192-193: I agree these sentences but according to the comment to Line 21, the effects of PAM treatments should be mentioned here.
Figure 6: I suggest to add the abbreviation of the treatment to all small figures for the easier understanding.
Discussion
Line 266: Please use Italics for scientific names of microbes.
Line 267: Please, delete “is” from this line.
Line 294: “However, these values were much higher in the current study.” The results on Fig. 5 and in Line 283 indicates lower values. Please, correct this sentence and check the next sentences, too.
Line320-321: Perhaps PAM is resistant to microbial degradation, but it increased soil respiration (Fig. 3). Perhaps, its effects you mentioned in the introduction could increase the microbial activity of treated soil?
Conclusions
Line 338: The PAM treatment effect on SR?
Author Response
Author Rebuttals to Initial Comments:
1 Overview of major changes
1.1 Introduction
In this section, we added the text that briefly describes the experimental area and provides information on the current soil status of the study site. The adverse effects of agricultural practices on soil quality were also mentioned (Line 31-39). The description of soil respiration, Q10 index (Line 67-86), and three experimental soil amendments (Line 64-65, 82-87) were also included in the introduction section.
1.2 Materials and methods
In this section, we added detailed information on the study site (Line 97-97), measurement of soil, and amendment properties, including sampling detail and soil properties and their measurement (Line 154-158). In addition, we also added a detailed description of data and model analysis (Line 178-185).
1.3 Results
We re-analysed the data shown in Table 2 and added a detailed description of the analysis in the result section (Line 227-238, 266-273).
1.4 Discussion
In this section, we provided an in-depth discussion on the effects of soil amendments on soil respiration and changes in soil respiration at different time points (Line 297-300, 310-313). Moreover, we discussed the effects of CEC and PAM amendment on SR (Line 356-360).
2 Detailed response to referees’ comments and suggestions
Response to Reviewer 1 Comments:
—General remarks: This is a well-designed and original work with detailed evaluation of results of a soil amendments test. I have only some remarks and suggestions to this work.
Response: Thank you for the positive evaluation of the manuscript and valuable suggestions.
—Line 21: “PAM amendment did not affect soil properties and SR.” Based on Fig. 3 a), different PAM treatments increased the SR comparing to the control, except in autumn.
Response: Thank you for bringing this important point to our notice. We apologize for not discussing it. In the revised manuscript, we have included this information, which reads as “PAM- amendment significantly increased SR in spring and winter compared to CK, PAM+ and PAM-W significantly increased SR, except in autumn; but PAM type and application method did not affect SR (Fig. 3b).”
—I suggest to add some sentences about soil respiration, as the main studied parameter of the work.
Response: Thank you for the constructive suggestion. We have incorporated the reviewer’s suggestion and mentioned brief information on soil respiration in the revised manuscript (Line 67-86).
—Line 129: Please, change “electronic” to electric.
Response: Thank you for pointing out the error. We have corrected it in the revised manuscript.
—Line 163: “Basic physicochemical properties of soil and amendments”
Response: Thank you for the constructive comment. We have corrected the error.
—Lines 170-173: Comparing the effects of BC and LD is a little bit confuse. First you mentioned no differences between them, after that difference in AN level was written, while in Table 2 AP and AN results are different between these two treatments.
Response: Thank you for raising this point. We have re-analyzed the data to address the reviewer’s query and modified the result section in the revised manuscript to “BC and LD amendments significantly increased EC, SOM, AN, AP, and CEC as compared to CK but did not affect TN, TP, and BD (Table 2). The pH, AN, AP, and CEC in BC treatment were significantly higher than LD treatment (P < 0.05), while LD had the highest EC. However, there was no significant difference between SOM, TN, TP, and BD in BC and LD treatments (P > 0.05). The PAM-, PAM+, and PAM-W amendments did not affect soil properties, except EC (Table 2).”
—Lines 192-193: I agree these sentences but according to the comment to Line 21, the effects of PAM treatments should be mentioned here.
Response: Thank you for the valuable suggestion. In the revised manuscript, we have included the effects of PAM treatments at this location, which reads as “PAM- amendment significantly increased SR in spring and winter compared to CK, PAM+ and PAM-W significantly increased SR, except in autumn; but PAM type and application method did not affect SR (Fig. 3b).”
—Figure 6: I suggest to add the abbreviation of the treatment to all small figures for the easier understanding.
Response: Thank you for the constructive suggestion. As suggested by the reviewer, we have modified the figures in the revised manuscript.
—Line 266: Please use Italics for scientific names of microbes.
Response: Thank you for pointing out the error. We have italicized names of microbes.
—Line 267: Please, delete “is” from this line.
Response: We have incorporated the reviewer’s suggestion.
—Line 294: “However, these values were much higher in the current study.” The results on Fig. 5 and in Line 283 indicates lower values. Please, correct this sentence and check the next sentences, too.
Response: Thank you for bringing this error to our notice. We have modified the sentence as suggested by the reviewer. The revised sentence reads “However, these values were much lower in the current study. It indicated that the temperature sensitivity of SR in the arid region cropland was lower than in another ecosystem.”
—Line320-321: Perhaps PAM is resistant to microbial degradation, but it increased soil respiration (Fig. 3). Perhaps, its effects you mentioned in the introduction could increase the microbial activity of treated soil?
Response: We agree with the reviewer. PAM application to soil significantly stimulated soil respiration in spring, summer, and winter. It indicated that PAMs could increase soil microbial activity. To address this point in the revised manuscript, we have included the following information “PAM application stimulated the SR in spring, summer, and winter. Similar findings were reported by Watson et al (2016). This study showed that PAM significantly increased SR as compared to control. Matsuoka et al (2002). demonstrated that Bacillus sphaericus and Acinetobacter strains could utilize PAM as the exclusive nitrogen and carbon source in the culture medium. In another study, the addition of wheat stubble with PAM enhanced PAM breakdown by basidiomycetes was demonstrated. This study showed that basidiomycetes could enzymatically degrade a myriad of nonphenolic complex substrates (Caesar-TonThat et al. 2008). It indicated that although the PAM was highly resistant to microbial degradation, its breakdown could increase the CO2 evolved from the microbial biomass.”
References
Watson, C.; Singh, Y.; Iqbal, T.; Knoblauch, C.; Simon, P.; Wichern, F. Short-term effects of polyacrylamide and dicyandiamide on c and n mineralization in a sandy loam soil. Soil Use Manage. 2016, 32, 127-136.
Matsuoka, H., Ishimura, F., Takeda, T., Hikuma, M. Isolation of polyacryla-mide-degrading microorganisms from soil. Biotechnology and Bioprocess Engineering, 2002, 7, 327–330.
Caesar-TonThat, T.C., Busscher, W.J., Novak, J.M., Gaskin, J.F. & Kim, Y. Effects of polyacrylamide and organic matter on microbes associated to soil aggregation of Norfolk loamy sand. Applied Soil Ecology, 2008, 40, 240–249.
—Line 338: The PAM treatment effect on SR?
Response: Thank you for raising this question. PAM treatment did not affect soil properties significantly, but it promoted soil respiration in spring, summer, and winter.
Reviewer 2 Report
The present manuscript entitled “Effects of Amendments on Physicochemical Properties and Respiration Rate of Soil from the Arid Region of Northwest China” is focusing on a very interesting and timing topic which is soil fertility restoration.
The Authors conducted a two-year field experiment to evaluate the effect of five soil amendments (biochar, leonardite, anionic polyacrylamide, cationic polyacrylamide powder, anionic polyacrylamide solution in water) on soil properties and respiration in the grey desert area (northwest China) compared to a control.
They observed that biochar and leonardite amendments altered the measured soil properties and respiration. Moreover, they found that climatic conditions affected soil respiration and temperature sensitivity of soil respiration.
Certainty, deeper analysis concerning how these soil amendments affected the soil microbial community composition and/or functionality could have further increase the interest in reading the manuscript.
The manuscript is interesting; however, it needs to be implemented and clarified as often it is difficult to follow. Moreover, I found incoherency in data presentation. Some measurements were performed on a monthly basis but data are reported sometimes at month level, sometimes at the season level, and some others as a single event. Finally, although authors found changes along time (as also mentioned in the abstract), this point is just mentioned and not very well discussed in the manuscript.
Following my comments:
Introduction
The introduction briefly describes the experimental area, the need to improve soil quality, and three soil amendments used in the experiment. However, it does not provide information on the current soil status in the study area and in which terms agricultural practices have negatively affected soil quality. Moreover, in the introduction, authors describe Polyacrylamide as a single soil amendment, but later they discern between anionic polyacrylamide, cationic polyacrylamide powder, and anionic polyacrylamide solution in water. A description of these three soil amendments in the introduction would increase the clarity of the manuscript. Moreover, why authors have chosen these 5 soil amendments? Authors should justify why they made this choice. Such justification can also provide the study hypothesis, which is absent in the current version of the manuscript.
In addition, Q10 index and carbon sequestration are not mentioned in the introduction, but, as two concepts are very important to follow the manuscript, I think they should be very well described in the introduction.
L36-38 Authors state “Soil amendments have been reported to be a highly effective approach for increasing the fertility and microbial activity of the soil” however not references are reported to support the statement. Please, provide references.
Material and methods
Study site: can you specify the soil type according to the international classification
L92: as there are more than one Polyacrylamide used in the experiment you can change PAM with PAMs
Measurement of soil and amendment properties: did you take all these measurements just “Before initiating the experiment”, as reported in L126? Or also after? If the latter when? Please, clarify this part
Statistical analysis: please, specify the model used for the analysis. Did the model consider the several measurements over the years?
Results
It is a bit difficult to follow the results without the Anova output. Please, provide it.
Table 2: please, include the data dispersion index.
Fig 3a is difficult to understand. In my opinion, it would be easier if authors can group bars of all the treatments by season. Moreover, why SR is reported two times? Figure 3’s caption is very confusing. Which one is the season in figure 3b?
L197: authors state: “The summer SR rate was significantly higher than spring, autumn, and winter. BC and LD amendments significantly increased SR rate as compared to CK, but no significant differences were observed between different amendments”. However, in fig 3 PAM+ and PAM-W seem to differ from the control as well. Moreover, Fig 3 shows that treatment differed also in spring, autumn, and winter, but these results are not reported in the text. Why?
Authors are presenting data by seasons and in table 3 Anova output by month. Please, use only one way to allow readers to understand results and increasing the manuscript readability.
Discussion
L247-254: Most of the parameters discussed in this part were not measured in the experiment (unless I did not get this part from the material and methods and the results sections). Authors can use references to explain their observed results, but the way this part is written seems that they have measured these processes. If so, please report the data and include this part in the results section.
L257-261: As above.
Conclusion
L340: I don’t understand this part. How carbon sequestration was improved? Can authors explain it?
Author Response
Author Rebuttals to Initial Comments:
1 Overview of major changes
1.1 Introduction
In this section, we added the text that briefly describes the experimental area and provides information on the current soil status of the study site. The adverse effects of agricultural practices on soil quality were also mentioned (Line 31-39). The description of soil respiration, Q10 index (Line 67-86), and three experimental soil amendments (Line 64-65, 82-87) were also included in the introduction section.
1.2 Materials and methods
In this section, we added detailed information on the study site (Line 97-97), measurement of soil, and amendment properties, including sampling detail and soil properties and their measurement (Line 154-158). In addition, we also added a detailed description of data and model analysis (Line 178-185).
1.3 Results
We re-analysed the data shown in Table 2 and added a detailed description of the analysis in the result section (Line 227-238, 266-273).
1.4 Discussion
In this section, we provided an in-depth discussion on the effects of soil amendments on soil respiration and changes in soil respiration at different time points (Line 297-300, 310-313). Moreover, we discussed the effects of CEC and PAM amendment on SR (Line 356-360).
2 Detailed response to referees’ comments and suggestions
Response to Reviewer 2 Comments:
—The present manuscript entitled “Effects of Amendments on Physicochemical Properties and Respiration Rate of Soil from the Arid Region of Northwest China” is focusing on a very interesting and timing topic which is soil fertility restoration.
The Authors conducted a two-year field experiment to evaluate the effect of five soil amendments (biochar, leonardite, anionic polyacrylamide, cationic polyacrylamide powder, anionic polyacrylamide solution in water) on soil properties and respiration in the grey desert area (northwest China) compared to a control. They observed that biochar and leonardite amendments altered the measured soil properties and respiration. Moreover, they found that climatic conditions affected soil respiration and temperature sensitivity of soil respiration. Certainty, deeper analysis concerning how these soil amendments affected the soil microbial community composition and/or functionality could have further increased the interest in reading the manuscript.
The manuscript is interesting; however, it needs to be implemented and clarified as often it is difficult to follow. Moreover, I found incoherency in data presentation. Some measurements were performed on a monthly basis but data are reported sometimes at month level, sometimes at the season level, and some others as a single event. Finally, although authors found changes along time (as also mentioned in the abstract), this point is just mentioned and not very well discussed in the manuscript.
Response: We are thankful to the reviewer for taking the time to review the manuscript and providing valuable and constructive feedback. We have provided a detailed response to each of the reviewer's comments, as mentioned below.
Following my comments:
—The introduction briefly describes the experimental area, the need to improve soil quality, and three soil amendments used in the experiment.
However, it does not provide information on the current soil status in the study area and in which terms agricultural practices have negatively affected soil quality.
Response: We agree with the reviewer. We have added details about the experimental area, soil, and amendments (Line 31-39).
—Moreover, in the introduction, authors describe Polyacrylamide as a single soil amendment, but later they discern between anionic polyacrylamide, cationic polyacrylamide powder, and anionic polyacrylamide solution in water. A description of these three soil amendments in the introduction would increase the clarity of the manuscript.
Response: Thank you for the suggestion. As suggested by the reviewer, we have added more details on PAMs in the introduction section, which reads as “Polyacrylamide (PAM), a synthetic soil conditioner, has been used since the 1990s to reduce soil erosion and enhance infiltration. PAM can be categorized into anionic (PAM-), cationic (PAM+), zwitterionic and non-ionic PAM (Sojka et al., 2007). Adverse effects of anionic PAMs on aquatic macrofauna, edaphic micro-organisms, or crop species have not been documented. Natural Resources Conservation Service (NRCS) has specified PAM- application for controlling irrigation‐induced erosion (Sojka et al., 2007).…Previous studies have rarely explored the application methods for PAM+ (Wang et al., 2020) and how these amendments' addition alter the grey desert soil’s properties. Besides, soil respiration in the arid region remains elusive.”
References
Sojka, R.E.; Bjorneberg, D.L.; Entry, J.A.; Lentz, R.D.; Orts, W.J. Polyacrylamide in agriculture and environmental land man-agement. Adv. Agron. 2007, 92, 75-162.
Wang, X., Yang, J., Zhang, S., Yao, R., Xie, W., Effects of different amendments application on cotton growth and soil properties in arid areas. Ecology and Environ-mental Sciences, 2020, 29(4): 757-762.
—Moreover, why authors have chosen these 5 soil amendments? Authors should justify why they made this choice. Such justification can also provide the study hypothesis, which is absent in the current version of the manuscript.
In addition, Q10 index and carbon sequestration are not mentioned in the introduction, but, as two concepts are very important to follow the manuscript, I think they should be very well described in the introduction.
Response: Thank you for the valuable remark. As per the reviewer’s suggestion, we have added more information on the selection of amendments, SR, and Q10 index in the introduction section (Line 56-58, 64-88).
—L36-38 Authors state “Soil amendments have been reported to be a highly effective approach for increasing the fertility and microbial activity of the soil” however not references are reported to support the statement. Please, provide references.
Response: Thank you for the suggestion. We have added this reference at the end of this sentence.
—Study site: can you specify the soil type according to the international classification
Response: The study site contained grey desert soil as per the Chinese soil classification system and calcareous desert soil as per the FAO soil classification (FAO, 2014).
References
Food and Agriculture Organization of the United Nations (FAO). World reference base for soils resources. World soil resource report No. 103. FAO, Rome, 2014.
—L92: as there are more than one Polyacrylamide used in the experiment you can change PAM with PAMs
Response: Thank you for the suggestion. We have modified the abbreviation.
—Measurement of soil and amendment properties: did you take all these measurements just “Before initiating the experiment”, as reported in L126? Or also after? If the latter when? Please, clarify this part
Response: Before initiating the experiment, the soil samples were collected from the experimental area from the diagonal direction to measure the basic soil properties as shown in Section 2.1 and Table 1. After recording the last SR measurement (May 2017), PVC collars were removed, and soil from 0~20 cm depth was sampled from each plot and then transported to the laboratory to analysed the basic properties of the soil.
—Statistical analysis: please, specify the model used for the analysis. Did the model consider the several measurements over the years?
Response: Yes, the model has considered several measurements over the years. We have mentioned the model used in the analysis and time in the result section of the manuscript, which reads as “Responses of SR to amendment, sample time (month), and their interaction was analyzed through two-way ANOVA. We used linear and non-linear models to quantify the correlation between environmental factors and SR (μmol/m2/s). Non-linear regression analysis of SR against ST, and linear regression analysis of Rs against AT (℃), ST (℃), SM (%) was performed. ” Also, we used the quadratic function model to determine the relationship between SR (μmol/m2/s) and SM (%).
—It is a bit difficult to follow the results without the Anova output. Please, provide it.
Table 2: please, include the data dispersion index.
Response: Thank you for the valuable remark. As suggested by the reviewer, we have provided the ANOVA output and dispersion index in the result section.
—Fig 3a is difficult to understand. In my opinion, it would be easier if authors can group bars of all the treatments by season. Moreover, why SR is reported two times? Figure 3’s caption is very confusing. Which one is the season in figure 3b?
Response: Thank you for this comment. We are sorry that we missed this important information. Fig. 3a and 3b represent the average rate of SR in different seasons and mean values (April 2015 to May 2017) of the different amendment treatments during the experimental period.
—L197: authors state: “The summer SR rate was significantly higher than spring, autumn, and winter. BC and LD amendments significantly increased SR rate as compared to CK, but no significant differences were observed between different amendments”. However, in fig 3 PAM+ and PAM-W seem to differ from the control as well. Moreover, Fig 3 shows that treatment differed also in spring, autumn, and winter, but these results are not reported in the text. Why?
Response: Thank you for bringing this important point to our attention. We have reported this information in the result section 3.2, which reads as follows SR exhibited clear seasonal patterns in different amendments. The highest and lowest respiration rates of all treatments were observed in June and January, respectively….. PAM- amendment significantly increased SR in spring and winter compared to CK, PAM+ and PAM-W significantly increased SR, except in autumn; but PAM type and application method did not affect SR (Fig. 3b).”
—Authors are presenting data by seasons and in table 3 Anova output by month. Please, use only one way to allow readers to understand results and increasing the manuscript readability.
Response: We have presented the data by months (Fig. 2) as well as by seasons (Fig. 3a). We measured the SR by months to investigate the temporal variation. To explore the seasonal differences in soil respiration, the data was represented according to the seasons (Fig. 3). Responses of SR to amendment, sample time (month), and their interaction was used in two-way ANOVA.
—L247-254: Most of the parameters discussed in this part were not measured in the experiment (unless I did not get this part from the material and methods and the results sections). Authors can use references to explain their observed results, but the way this part is written seems that they have measured these processes. If so, please report the data and include this part in the results section.
—L257-261: As above.
Response: Thank you for your suggestions. In this part of the discussion, we refer to the experimental data of other researchers to support our experimental results. As our study is not an integrated analysis, we believe that it is out of the scope of the current study to collect relevant research results data.
—L340: I don’t understand this part. How carbon sequestration was improved? Can authors explain it?
Response: If the application of soil amendments increases soil organic matter content and stability and reduces soil respiration rate and respiration temperature sensitivity coefficient, we believe that soil amendments can improve soil carbon sequestration. However, in our research, biochar and leonardite increased soil organic matter and soil respiration but reduced the temperature sensitivity coefficient of soil respiration. Thus, we argue that biochar and leonardite cannot negate the net carbon accumulation by increasing carbon loss through SR in cropland grey desert soil.
Reviewer 3 Report
Dear author,
I have added a number of comments and suggestions throughout the text. Please place emphasis on improving the figures I have indicated in the text.
Kind regards

Author Response
Author Rebuttals to Initial Comments:
1 Overview of major changes
1.1 Introduction
In this section, we added the text that briefly describes the experimental area and provides information on the current soil status of the study site. The adverse effects of agricultural practices on soil quality were also mentioned (Line 31-39). The description of soil respiration, Q10 index (Line 67-86), and three experimental soil amendments (Line 64-65, 82-87) were also included in the introduction section.
1.2 Materials and methods
In this section, we added detailed information on the study site (Line 97-97), measurement of soil, and amendment properties, including sampling detail and soil properties and their measurement (Line 154-158). In addition, we also added a detailed description of data and model analysis (Line 178-185).
1.3 Results
We re-analysed the data shown in Table 2 and added a detailed description of the analysis in the result section (Line 227-238, 266-273).
1.4 Discussion
In this section, we provided an in-depth discussion on the effects of soil amendments on soil respiration and changes in soil respiration at different time points (Line 297-300, 310-313). Moreover, we discussed the effects of CEC and PAM amendment on SR (Line 356-360).
2 Detailed response to referees’ comments and suggestions
Response to Reviewer 3 Comments:
—L39, L50, and L260: add references
Response: Thank you for the constructive inputs. We have added references for the statements as suggested by the reviewer.
—L39: The author should write the names of cities or areas using brackets, the author must understand that not all people who will read this work know that area.
Response: Thank you for the valuable suggestion. In the revised manuscript, we have mentioned the cities name in the brackets and revised the sentences to “Leonardite (LD), a waste product of coal production that is commonly present on the surface of coal, is widely distributed across Xinjiang at Hutubi County, Shanxi (Luliang City), Inner Mongolia (Ordos) provinces in China.”
—L60: the author did not previously explain these abbreviations
Response: Thank you for pointing this error. We have expanded the abbreviation as suggested by the reviewer.
—L73: gps coordinates
Response: We have added GPS coordinates in Line (L95).
—L80: What part of the cotton plant was pyrolyzed?
Response: We have used cotton talk for biochar preparation. The method is described in the revised manuscript as “The biochar used in this study was produced using a cotton stalk, where cotton was pyrolyzed with a residence time of 2 h at 400 °C in a muffle furnace and later grounded to powder.”
—L109-110: the author should include the name of the equipment he/she has used. type? references?
Response: Soil temperature at 5 (ST5), 10 (ST10), and 15 (ST15) cm depth were measured using 6 groups of geothermometers (Wuqiang Inc., Hengshui, Hebei, China).
—Table 1: statistical analysis comparing the differences between the original amendments, otherwise, how will he know if the effect produced in the soil is really due to an amendment?
Response: Thank you for raising this point. We have added the statistical comparison of the differences between the amendments.
— L167 3.1. Subsection
Response: Thank you for pointing out the error. The secondary heading should be the “Effect of soil amendments on soil properties.”
— Figure 1. what is this treatment? These graphics must be much improved, you can't see anything at all.
Response: Since the different experimental treatments were conducted in the same area, recorded air temperature did not represent a particular treatment. As the data of soil moisture for different treatments lies closely, it is difficult to clearly demarcate the different data points. Thus, we placed a small picture in Figure 1c to show that there is no significant difference in soil water content between different treatments.
— Table 4 the author presents a lot of tables and figures but hardly comments on the results. The author should give more detail on the results obtained.
Response: Thank you for the suggestion. We have incorporated the reviewer’s suggestion in the revised manuscript and added the following information on the data in result Section 3.3 “For CK and BC treatment, the two-variable model could be used to explain the correlation of SR to AT and SM with R2 ranging from 0.873 to 0.896 and significant regressions (P < 0.0001). The SR of LD, the one-variable linear model, could be used to describe the correlation between SR and ST with 0.722 R2. SR of PAMs could be predicted by ST and SM. The equations explained 64.4–74.6% of the variation in SR in our data (Table 4).”
—L 252 this contradicts many previous studies..... the author needs to better substantiate this claim.
Response: We appreciate this suggestion by the reviewer. We have discussed the contradiction in the discussion section of the manuscript, which reads as “Biochar amendment increased the labile exogenous organic carbon in the soil. Labile fractions of biochar are an important source of carbon that is used as a selective substrate in microbial activity (Sagrilo et al., 2014). Smith et al. (2010) suggested that soil CO2 is primarily contributed by the decomposition of labile carbon fractions of the biochar.”
References
Sagrilo, E.; Jeffery, S.; Hoffland, E.; Kuyper, T.W. Emission of CO2 from biochar amended soils and implications for soil organic carbon. GCB Bioenergy 2014, 7, 1294-1304.Smith, J.L.; Collins, H.P.; Bailey, V.L. The effect of young biochar on soil respiration. Soil Biol. Biochem. 2010, 42, 2345–2347.
—L 303 the author should not repeat the results data in the discussion section.
Response: Thank you for the suggestion. As suggested by the reviewer, we have deleted the repetitive result data from the discussion section.
—L 316 the author should rely on exchange cations and base saturation rather than functional groups .....
Response: Thank you for this very useful comment. We have added a more discussion on the impact of biochar on CEC as follows: “Biochar and leonardite application increased soil CEC value as compared to CK. Liang et al (2006). demonstrated that CEC increase could be attributed to the high surface area and charge density of the biochar. In addition, oxidation of aromatic carbon on the biochar surface to carboxylic groups increased the CEC (Mikutta et al., 2005; Peng et al., 2011).”
References
Liang, B., Lehmann, J., Solomon, D., Kinyangi, J., Grossman, J., O’Neill, B., Skjemstad, J.O., Thies, J., Luizao, F.J., Petersen, J., Neves, E.G., 2006. Black carbon in-creases cation exchange capacity in soils. Soil Science Society of America Journal 70, 1719–1730.
Mikutta, R., Kleber, M., Kaiser, K., John, R., 2005. Review: organic matter removal from soils using hydrogen peroxide, sodium hypochloride, and disodium perodisulfate. Soil Science Society of America Journal 69, 120–135.
Peng, X.; Ye, L.; Wang, C.; Zhou, H.; Sun; B. Temperature and duration-dependent rice straw-derived biochar: Characteristics and its effects on soil properties of an ultisol in southern China. Soil Tillage Res. 2011, 112, 159-166.
Round 2
Reviewer 2 Report
The manuscript has been markedly improved and, in my opinion, it is now suitable to be published in the journal.
Best regards
Reviewer 3 Report
Dear author,
I believe that with the changes made, the manuscript has been improved.
Kind regards